# The Breakthroughs and Caveats of Using Human Pluripotent Stem Cells in Modeling Alzheimer’s Disease

**DOI:** 10.3390/cells12030420

**Published:** 2023-01-27

**Authors:** Katja Maria Sahlgren Bendtsen, Vanessa Jane Hall

**Affiliations:** Group of Brain Development and Disease, Department of Veterinary and Animal Sciences, Faculty of Medical Sciences, University of Copenhagen, Dyrlægevej 100, 1870 Frederiksberg, Denmark

**Keywords:** induced pluripotent stem cells, Alzheimer’s disease, disease modelling, neurons, astrocytes, microglia, drug discovery, APP processing, tau phosphorylation

## Abstract

Modeling Alzheimer’s disease (AD) using human-induced pluripotent stem cells (iPSCs) is a field now spanning 15 years. Developments in the field have shown a shift in using simple 2D cortical neuron models to more advanced tri-cultures and 3D cerebral organoids that recapitulate more features of the disease. This is largely due to development and optimization of new cell protocols. In this review, we highlight recent major breakthroughs in the AD field and the implications this has in modeling AD using iPSCs (AD-iPSCs). To date, AD-iPSCs have been largely used to recapitulate and study impaired amyloid precursor protein (APP) processing and tau phosphorylation in both familial and sporadic AD. AD-iPSCs have also been studied for varying neuronal and glial dysfunctions. Moreover, they have been useful for discovering new molecular mechanisms, such as identifying proteins that bridge APP processing with tau phosphorylation and for identifying molecular pathways that bridge APP processing dysfunction with impaired cholesterol biosynthesis. Perhaps the greatest use of AD-iPSCs has been in discovering compounds via drug screening, that reduce amyloid beta (Aβ) in neurons, such as the anti-inflammatory compound, cromolyn, and antiparasitic drugs, avermectins. In addition, high content screening using AD-iPSCs has led to the identification of statins that can reduce levels of phosphorylated tau (p-Tau) in neurons. Some of these compounds have made it through to testing in human clinical trials. Improvements in omic technologies including single cell RNA sequencing and proteomics as well as advances in production of iPSC-cerebral organoids and tri-cultures is likely to result in the further discovery of new drugs and treatments for AD. Some caveats remain in the field, including, long experimental conditions to create mature neurons, high costs of media that limit research capabilities, and a lack of reproducibility using current iPSC-cerebral organoid protocols. Despite these current limitations, AD-iPSCs remain an excellent cellular model for studying AD mechanisms and for drug discovery.

## 1. Introduction

Since the emergence of human-induced pluripotent stem cells (iPSCs) in 2007 [1], over 600 articles have been published on PubMed which include the terms “pluripotent stem cells” (PSCs), “modeling” and “Alzheimer’s disease” (AD). This highlights an extensive amount of research in this field over the past 15 years. In particular, the hope of studying AD using iPSCs (AD-iPSCs) was first to determine if these models could replicate the features of the disease in vitro. However, following protocol optimizations and developments over the years, AD-iPSCs are being increasingly used to learn more about the disease mechanisms and to identify compounds for treating the disease. We first highlight the latest findings in AD and discuss the implications of these on modeling the disease using AD-iPSCs. We then review the best and most recent research breakthroughs in the AD-iPSC field and highlight the current challenges that are being faced. Together, this review highlights the most recent research breakthroughs in the AD field and provides an overview of the progress in modeling and learning about the disease using iPSCs. 

## 2. The Latest New Findings on Pathology and Treatment of AD

There have been several noteworthy discoveries on AD in the last decade. New forms of AD are constantly being discovered including late-onset sporadic forms with similarities to AD and early-onset familial forms. One of the most important is the characterization of a related dementia, limbic-predominant age-related TDP-43 encephalopathy (LATE). LATE neuropathological changes include characteristic restricted distribution of TDP-43 proteinopathy, often associated with amyloid beta (Aβ) plaques and tauopathy [2]. LATE-NC is now believed to be a common dementia in the very old (>80 years) with approximately 20% of individuals affected [2]. It is characterized as a dementia of the Alzheimer’s type with genetic risk factors that overlap both frontotemporal dementia (FTD) and AD (*APOE, GRN, TMEM106B, ABCC9, KCNMB2*) [2]. Another familial AD has also been reported and has been named the Uppsala APP deletion [3]. This familial AD results in an aggressive form of AD, beginning in individuals in their early 40s [3] (Figure 1A). These newly characterized forms of AD increase the complexity in our understanding of the disease and highlights a diversity in AD and dementias that likely stems from many mutations that we continue to identify (Figure 1A). Genome-wide associate studies (GWAS) have made great progress in recent years in finding single nucleotide polymorphisms that are associated with AD, strengthening the notion that genes play an important role in disease onset. Two new genes were recently discovered, including *S100B* and *CD44*, as both have dysregulated long coding RNAs clustered near their loci in AD patients [4]. The largest GWAS study to date on AD was published a year later, in 2021. This study included over 1 million individuals and resulted in 38 gene loci associated with AD being identified, many of which have already been identified [5]. However, five of these, *AGRN*, *TNIP1*, *HAVCR2*, *NTN5*, and *LILRB2* have not previously been associated with dementia [5] (Figure 1B). In sum, new genes have been added to a growing and long list of genes. All appear to play a role in AD, indicating that a multifactorial mutational genetic component gives rise to diverse symptomatic and pathological profiles. However, in the clinic, current diagnostic tools and cognitive tests diagnose the patients according to standard criteria and as having the same disease, which may in fact be untrue. Another noteworthy finding has been the characterization of the different structures of Aβ42 filaments which identifies differing structures in familial versus sporadic AD. Cryo-electron microscopy has identified two forms of Aβ42 filaments [6]. Type 1 is found in sporadic AD brains whilst type 2 is found in familial forms of AD and both differed to in vitro forms of Aβ42 [6] (Figure 1C). This could have important implications in how clinicians end up treating these different forms of AD. In addition, can in vitro studies replicate Aβ pathology well-enough if different fibril forms of Aβ42 are produced? The implications of differences in fibril structure needs further exploration.

Research focus has expanded from studying the two major hallmarks of the disease, amyloid plaques and tau tangles, to also studying organelle dysfunction within neurons and effects on the blood vasculature. Regarding organelle dysfunction, mitochondria appear particularly vulnerable. Mitophagy has been recently found to be impaired in AD patient hippocampal neurons by as much as a third to half of normal age-matched healthy brains [7]. In addition, accumulated, damaged mitochondria that are smaller in size have been identified in AD hippocampal neurons [7]. These neurons are less effective at producing energy to support important cellular functions. More research is also emerging in the role of the blood vasculature in AD, with clear evidence showing how cerebral blood flow, blood vessels and the blood–brain barrier (BBB) is impaired in patients [8]. Even the BBB has been found to be impaired in patients at risk of developing AD [9]. Microvascular changes have been reported in AD patient brains, such as narrowing of the capillary beds [10]. Vascular dementia has long been regarded as the dementia with impaired blood vessels, however, researchers are now aware that vascular impairment is also found in other dementias, including AD. Together, these studies highlight how constricted vasculature might limit access of the brain to nutrients, or limit the ability to carry away toxic amyloid. Neurons themselves may not function well due to internal organelle dysfunction due to restricted access to nutrients the blood supplies. 

Research on the role of inflammation, gut microbiome, an overactive immune system, and infection in AD has also increased. The activation of complement has long been understood, but new studies have increased the understanding on the diversity of complement proteins involved in AD and their role on synapse health and neurodegeneration. For example, too much of the complement protein C1Q together with microglia resulted in excessive synapse pruning in mouse models of AD [11]. Recently, another complement protein, C3, was found in human AD patients around neuronal synapses and in the CSF [12]. This protein has also been shown to play a major role in synapse loss and neurodegeneration in mouse models of AD [12]. The role of infection in AD onset is also emerging as a risk for AD. A recent epidemiological study highlights how any type of hospital-treated infection occurring in early to mid-life is associated with the development of AD in patients that are diagnosed prior to 60 years of age [13]. This is the most convincing study to date correlating infection with AD. Together, these studies highlight how peripheral infections play an important role in disease onset and that the immune system has an important role. Exact mechanisms that connect the role of infection and onset of AD however remain to be discovered. Alterations to the microbiome have also been detected in AD patients, that are not observed in patients with mild cognitive impairment (patients with early AD symptoms) [14]. Particularly, a lower microbiome diversity has been reported [14] and genetics, such as bearing of different APOE isoforms appear to modulate microbiome diversity [15]. Further, different levels of specific microbiome species colonizing the gut correlate with pTau and amyloid status [16]. What remains unknown is whether AD induces these observed changes in the gut microbiota, or if changes to the microbiota trigger mechanisms that induce the disease.

Specific cell types and cell subtypes have also been found to be affected differently in AD. The emergence of single cell RNA sequencing (scRNA-seq) over the last few years has allowed researchers to explore the heterogeneous effects of AD on cell types. For example, one study on human AD patient brains resulted in the identification of a vulnerable glutamatergic neuron expressing the gene, *RORB* in the entorhinal cortex. This part of the brain is affected early in the disease [17]. In addition, astrocyte populations in the entorhinal cortex have been shown to be reduced in AD patient brains [18] confirming that not only neurons but glia are affected in certain brain regions. A particular subset of microglia has also been identified as being affected by AD and is reduced in numbers [19]. The outcomes of these studies help highlight that it may be necessary to produce and study these newly identified vulnerable and relevant cell subtypes in vitro from iPSCs to learn more about the disease. 

Two new monoclonal antibody-based drugs show excellent clearance of amyloid from the brain which have moderate effects in patients. Both have reached the US market [20]. Aducanumab (marketed as Aduhelm), when used at high doses, has shown a slow-down in cognitive decline by 30% in a Phase III clinical trial and has been approved by the FDA in the US for patients with mild dementia or mild cognitive impairment [21]. This antibody targets and clears aggregated forms of Aβ and has also been shown to reduce pTau [22]. The second drug is Lecanemab (marketed as Leqembi), which binds to soluble Aβ protofibrils (Figure 1D). However, the latest findings from a phase III clinical trial shows promising results, with a slowing of cognitive decline by 27% in patients following 18 months of treatment [23,24]. One cautionary note is the potential side-effect in swelling of the brain with both drugs. Amyloid-related imaging abnormalities (ARIA) have been observed in 12.6% of patients taking Lecanemab during the newly reported Phase III clinical trial [23] and in a third of patients taking Aducanumab. Microbleeds were also an occurrence and a quarter of all patients taking Lecanemab experienced infusion-related reactions [24], with some deaths also reported [20]. Therefore, investigations of the safety of both Lecanemab and Aducaumab continue, and other countries await further results before considering their approval. Further, both Lecanemab’s and Aducanumab’s fast track release onto the US market has resulted in split opinions between clinicians amid these safety concerns and doubts of possible benefits for patients [20,21]. Despite these concerns, both drugs are considered large breakthroughs for the treatment of AD and provide hope for patients and families alike. Another drug has received some attention for its promise in treating AD. A marine-derived oligosaccharide (compound GV-971) has been tested in Phase III clinical trials and has been approved for treatment of AD in China for patients with mild to moderate AD [25]. This came as a surprise for the medical community since little evidence has been disclosed on its efficacy [26]. Previous studies, have shown GV-971 acts primarily in the gut, altering the microbiome of patients, and by doing so, reducing both infiltration of immune cells into the brain and neuroinflammation [27]. This compound has been shown to destabilize Aβ fibrils and prevent their formation in cellular models [28] and improves memory impairment in a scopolamine-induced amnesia rat model [29]. No direct reports have been performed on AD models. Another natural compound has raised interest in the AD medical community. The pigment found in turmeric, curcumin, appears to be a promising treatment for AD. Curcumin was able to reduce amyloid and neuroinflammation in an APP^swe^ transgenic mouse model after six months of treatment, as well as decrease Aβ secretion in cultured human brain slices [30]. Phase I clinical trials in humans have so far revealed variable outcomes [13], with gastrointestinal adverse effects being reported as the most adverse symptom. However, some clinical trials are still ongoing, and hope remains that dosages can be optimized for the most promising outcome. Together, these last few years have shown promising advances in the development of new treatments for patients that remove toxic amyloid and clinical trials continue for other promising compounds. 

It is important that this new knowledge acquired in the AD field is recapitulated in future AD-iPSCs studies. However, an increasing complexity in our knowledge of AD means that the disease becomes more challenging to study in vitro using simple cellular models. New advances in AD research require refinements of research questions when using iPSCs. One caveat with this approach is that stem cell research remains one step behind discoveries in the field. On the other hand, modeling AD with AD-iPSCs allows researchers to study certain pathological aspects within key affected cells in the disease in detail. This allows for new discoveries to emerge from the iPSC models themselves. Studies using AD-iPSC models have mostly focused on neuronal pathology, since cortical neurons have been the main cell type produced from iPSC. However, new advances are now allowing researchers to better model the complex brain environment, which is propelling the field forward. We now focus the rest of the review on iPSC models and how they have contributed to our understanding of AD. 

## 3. An Overview of iPSC Studies Modelling AD

Advances in production of cell types from pluripotent stem cells (PSCs) now allow for several neural cell types to be produced using well-validated, reproducible protocols. These include postmitotic cortical excitatory neurons [31,32,33], hippocampal neurons [34], basal forebrain cholinergic neurons [35,36], GABAergic inhibitory neurons [37], astrocytes [38,39] and microglia-like cells [40,41,42,43,44,45]. It is important to note, however, that researchers should consider which cell subtypes are most relevant for studying AD. Most studies to date have focused on studying cortical neurons in 2D cultures. Emerging evidence suggests for example that *RORB*-expressing neurons from the entorhinal cortex are particularly vulnerable and found in a part of the brain first affected by the disease [17]. The choice of subtype to study should be taken into consideration, as well as whether new protocols should be developed to produce more vulnerable neuronal cell types. Recently, our group developed a protocol to produce a principal entorhinal neuron from human iPSCs, resembling stellate cells by using a forward programming approach [46]. Stellate cells expressing Reelin are one of the first cell types to accumulate Aβ intracellularly [47]. These cells also reside in LII which is a cellular layer most heavily affected in the entorhinal cortex in AD [48]. Our own scRNAseq analyses of the entorhinal cortex show a cell population we identified as stellate cells also expresses *RORB* [46], further highlighting stellate cells are a useful cell type for studying AD. 

There has been a transition in modeling diseases using iPSCs from using simple culture conditions (Figure 2A) to more complex culture conditions. In general, differentiated iPSCs are cultured as adhered cells in dishes, termed two dimensional (2D) cultures. However, over the past few years, there have been advances in creating more complex systems to recreate AD by using multiple cell cultures either in 2D (co- or triple-cultures) [41,49], or by using cerebral organoids cultured in suspension in three dimensions (3D) [50,51,52,53], referred to more simply as “organoids” herein (Figure 2B). These organoids have several cell types and may self-organize from residing progenitors within them. There may be great advantages in using complex cell cultures in recapitulating the disease in vitro. Complex cultures help resemble the complexity of the brain parenchyma which we know also includes dysregulated microglia, astrocytes, and vasculature in AD. The addition of astrocytes, microglia and modulating exposure to a dysfunctional and leaky BBB will help to better recapitulate more features of the disease.

## 4. How Well Do iPSCs Recapitulate Major Features of AD Disease?

Here, we summarize the major pathologies of AD that have been recapitulated across selected studies from the literature using human iPSCs derived from AD patients or iPSCs containing CRISPR-engineered/gene-edited AD mutations (See Figure 2C). 

### 4.1. Amyloid Beta (Aβ)

Of all the pathologies in AD, iPSC models have been used to predominantly study the processing of APP and secretion of different Aβ species. Human iPSC-derived neuronal and organoid cultures from both familial and sporadic AD lines show convincingly that they can model increased secretion of Aβ and more toxic and longer forms of Aβ than isogenic and healthy control iPSC lines. See Table 1. Fewer studies have been able to recapitulate deposition of extracellular plaques in vitro. However, two studies using organoids from both familial AD and sporadic AD exposed to serum (which mimics the exposure of the brain tissue to a leaky BBB) have recently been able to demonstrate deposition of extracellular Aβ [54,55]. One study, which has carefully analyzed Aβ from several familial iPSC-cortical neurons with different APP and PSEN1 mutations, concluded that the only consistent change in abundance of secreted Aβ species across the iPSC lines was the increased secretion of Aβ42 [56]. There was no consistency in change of total Aβ or Aβ42:40 [56]. This is also evident in Table 2 when comparing levels of Aβ species across several familial and sporadic AD lines, although not all AD-iPSC lines from familial and sporadic backgrounds show increased secretion of Aβ42. This is likely a reflection of differences in APP processing that differs between the numerous mutations and risk genes associated with AD [57,58]. One powerful study conducted by Lagomarsino and colleagues examined the correlation of AD pathology in late-onset sporadic AD-iPSC-cortical neurons compared to brain and omics data from the same patients [59]. This was performed with 53 patients of which 53 iPSC lines were generated. This is the best study to date that correlates the pathology of a large number of iPSC lines with detailed records from patient postmortem brains and RNA sequencing data obtained from the patient’s prefrontal cortex. Correlations were most convincing between intracellular Aβ42:40 from the cell lysate of iPSC neurons and the number of observed neuritic plaques found in the patient’s brains. Interestingly, secreted levels of Aβ from the iPSC neurons did not correlate very well to levels in the brain. This suggests that evaluating intracellular loads of Aβ should be considered as a more useful way of replicating patient data when evaluating iPSC neurons. One of the most striking outcomes from this study was the supporting evidence for a molecular cross link between impaired APP processing and disrupted phosphorylation of tau (pTau). The study reported that altered APP processing in the AD-iPSC-cortical neurons led to a reduction of the serine/threonin–protein phosphatase PP1-alpha catalytic subunit (PPP1CA). The study also showed that a reduction of protein phosphatase 1 (PP1) led to an increase in expression of major pTau in the neurons [59]. PP1 and other protein phosphatases including PP2A, PP2B, and PP5 are known to phosphorylate tau and have previously been linked in playing a role in AD [60]. In another comprehensive study, the secretion of Aβ42 was found to correlate negatively with Aβ42 taken from the same patient’s cerebral spinal fluid (CSF) [61]. This is thought to be due to sequestration of non-Aβ42 in non-soluble plaques that remain in the brain. However, the levels of secreted Aβ40 and Aβ38 correlated closely with levels found in the CSF [61]. One testimony to the caution of measuring secreted Aβ subunits comes from a paper that highlighted a large variability in the secretion of Aβ43, 42, 40, and 38 species over time within iPSC lines derived from familial AD patients [62]. It was only when ratios were analyzed to the most abundant secreted product, Aβ40, that consistent findings could be found [62]. Comparisons of iPSC-derived Aβ38:40 was the most consistent measurement that compared well with CSF and brain homogenate from the same patients and Aβ42:40 best with soluble fractions of brain cortical tissue, but not with CSF levels [62]. More variability in the level of APP processed subunits were observed in 3D cultured organoids from familial AD patients, but again, Aβ42:40 ratios remained consistent [62]. Together, these studies highlight that intracellular levels of Aβ42:40 are an excellent correlator to patient amyloid plaques in the brain. Secreted amounts of Aβ species vary considerably and may depend on processing differences associated with the mutation of interest. In addition, iPSC studies have been able to test and validate particular hypotheses that link impaired APP processing with hyperphosphorylation of tau which help to forge the molecular bridge between these two pathologies. 

### 4.2. Tau

The next most investigated pathology in AD-iPSC models has been the study of tau phosphorylation. The investigation of tau in iPSC cultures has also led to interesting findings which indicate that the level of pTau may not replicate the level of pTau in patient brains as well as hoped. The study by Lagomarsino and colleauges did not find an especially good correlation in the levels of pTau produced from iPSC-cortical neurons compared to the level of pTau found in the corresponding patient brains. In fact, a negative correlation was observed when comparing the quantity of the major band of pTau and observed tangles in the brain. One exception was found where the levels of the commonly-used AT8 antibody (which measures p-total tau) in the iPSC cortical neurons correlated well with the major tau levels in the human brain samples. However, this correlation was only observed when measuring total pTau by using Western blotting and not from correlations with the number of tangles. However, increased levels of pTau have been observed in most AD-iPSC models that have investigated tau phosphorylation (Table 2). We report here only two studies that did not detect an increase in pTau from iPSC sporadic organoids compared to controls. Interestingly, the extracellular deposition of tau as tangle-like morphology has only been reported in one iPSC paper to the best of our knowledge. Here, neurofibrillary tangles were observed in PSEN iPSC organoids [69].

### 4.3. Other Proteinopathies

Other proteinopathies have been described in AD but have not been well studied in AD-iPSC models. Lewy bodies are protein accumulations mostly associated with Parkinson’s disease (PD), but are also cohabiting in AD patient brains [77]. In fact, up to 60% of familial AD and sporadic AD brains contain Lewy bodies specifically in the amygdala [78]. Given Lewy body anatomy in AD differs to PD [78], more studies are required to study this pathology hallmark associated with AD [79]. In the case of AD-iPSCs, production of amygdala-specific excitatory and inhibitory neurons would be required to examine AD-related Lewy body pathology, of which we lack protocols for. Another proteinopathy described in LATE AD is the accumulation of the protein TDP-43. Some initial steps have been taken to investigate TDP-43-related pathology using iPSCs. IPSC lines derived from behavioral variant FTD (bvFTD) patients have been used to study changes in gene expression in patient cell lines [80]. Another recent study has shown stathmin-2 (STMN2) is associated with TDP-43 inclusions using human iPSCs [81]. However, to the best of our knowledge, no research has studied TDP-43 dysregulation in the context of AD, since iPSCs have not been generated from LATE AD patients. Another pathology described in AD are Hirano bodies, which are made of actin-rich paracrystalline inclusions [82]. Given these are also found in dementia patients, ALS patients and to a lesser degree even healthy persons [82], this pathology also remains overlooked in AD-iPSC models. In sum, we have much more to learn from iPSC models of AD and should consider these less understood proteinopathies associated with AD. 

### 4.4. Organelle Pathology 

Microscopic and molecular studies on AD-iPSC models have increased our understanding of organelle dysfunction in AD. Mitochondrial dysfunction, decreased autophagy, decreased mitophagy, and large early endosomes have been reported (See Table 3 and Figure 2). Mitochondria from AD-iPSC neurons are reported to have reduced mass, have decreased mitochondrial membrane potentials, and altered respiration (See Table 3). However, reported levels of ROS and superoxide levels appears to vary between cell lines [83,84,85] (Table 3) despite increased ROS being a common feature in AD [86]. Increased ROS in AD is attributed to a deficiency of cytochrome c oxidase (COX) [86] which has also been reported in iPSC cortical neurons and organoids [85]. The clearance of damaged mitochondria via mitophagy is also impaired in iPSC neurons [7]. Interestingly, a comparative study of iPSC astrocytes with iPSC neurons indicated that astrocytes had fewer mitochondrial dysfunctional features but did have increased mitochondrial membrane potentials [85]. One iPSC brain organoid study was able to determine a direct molecular correlation with mitochondrial dysfunction and APP processing. Following knockout (KO) of the mitochondrial peptidase PITRM1, an increase occurred in Aβ in iPSC neurons [87]. This also resulted in protein aggregation, tau pathology, and neuronal death in the organoids [87]. This research strengthens the potential interrelationship of genes in mitochondrial function that can induce AD-specific pathology. An enlargement of early endosomes has been reported in several studies in iPSC neurons (Table 3). Interestingly, these defects have not been observed in iPSC microglia-like cells [88]. This impairment of endosomal trafficking is thought to be due to impaired APP processing and trafficking, since cleavage of APP occurs in early endosomes [89]. However, increased numbers of large endosomes have also been reported in iPSC organoids and decreased autophagy observed in iPSC neurons (Table 3). The endoplasmic reticulum (ER) is an important trafficking organelle that plays a vital role in protein folding, which was also shown to be dysregulated in some, but not all studied iPSC-AD lines [64] (Table 3). Together, iPSC studies have been able to largely recapitulate organelle dysfunction in neurons observed in AD patient brains. Further studies have been able to study molecular pathways that couple organelle dysfunction to impaired APP processing. Improving mitochondrial function could be performed using antioxidant drugs. Antioxidants such as Vitamin E, caffeine, and turmeric have been well acknowledged as potentially having an impact on delaying disease progression or disease onset [90]. Interestingly, an iPSC neuron study using a healthy patient background showed that Vitamin E deficiency can induce ER stress and oxidative stress phenotypes [91], similar to pathologies observed in AD-iPSC-derived neuron studies. However, Vitamin E is a common component in stem cell media, which may circumvent potential deficiencies observed in patients, but may also rule out Vitamin E deficiencies being responsible for oxidative stress or ER stress reported in cell lines.

### 4.5. Neuronal Dysfunction

Studying AD-iPSC neurons in vitro has highlighted neuronal dysfunction, however variable reports in some measurements such as neuronal excitability have been reported (Figure 2c and Figure 3). Decreased neurite lengths and decreased dendritic branching have been reported by more than one study (Table 4). This phenotype appears to reflect an altered cytoskeleton and reflect dystrophic dendrites that have been reported in AD [93]. Of all neuronal dysfunctions, synapse loss is the strongest indicator of dementia severity in AD [94]. Only a few studies have investigated synapses in AD-iPSC neurons. Studies have reported reduced synapses in iPSC organoids treated with serum [55] and in iPSC cortical neurons [95], whereas another study reported increased number of synapses in sporadic AD induced neurons (iNeurons) [72] (Table 4). The notion of exposing organoids to serum is thought to mimic the breakdown of the BBB and exposure to blood plasma and this exposure indeed exacerbated AD pathology, with increased Aβ and pTau observed [55]. Given glia play an important role in pruning synapses, co-cultures of neurons and microglia may better evaluate this phenotype. This has been highlighted already in one study which reported that Down syndrome iPSC microglia-like cells excessively prune neurons [96]. A second study using iPSCs has studied increased synapse pruning in microglia-like cells derived from schizophrenia patient iPSC on cultured neurons [97,98]. However, no such studies have yet been conducted on microglia derived from common familial mutations or in sporadic AD-iPSCs. It is considered that cholesterol secretion is particularly important for neuron synapse function and for cholesterol deposition in the plasma membrane. This is a particularly significant cell process in the case of sporadic AD, where impaired cholesterol signaling has been attributed to the risk gene APOE4 [99]. Both the plasma membrane and ER are important in cholesterol signaling since this is where the protein machinery is located for regulating cholesterol synthesis. Cholesterol synthesis has been well studied in a couple of AD/APOE4 iPSC studies (Table 5). Increased cholesterol biosynthesis has been reported in iPSC organoids, but pathology has been better studied in iPSC astrocytes (see Section 4.6).

Altered excitability of AD-iPSC neurons has also been reported with some studies indicating hyperexcitability and others reporting decreased excitability (Table 4). In the case of using iPSC organoids, a cautionary note of advice comes from one study which highlighted a large variability in the cell types found within cultured iPSC organoids derived from the same parental line and also a large variation in neuronal activity between organoids [100]. The reported variabilities may indicate differences exist between different AD mutations and/or that variability exists in iPSCs due to in vitro factors. Follow up studies are required to carefully evaluate differences between mutations to rule out in vitro effects that may compromise pathology read-outs. 

### 4.6. Glia Dysfunction

More attention is being placed on the role of glia in AD pathology and some AD-iPSC studies have studied the pathology and features of AD-iPSC microglia-like cells and astrocytes in vitro (Table 5). Molecular connections between perturbed cholesterol synthesis and impaired Aβ have been studied particularly in sporadic AD associated with the risk gene, APOE4. One iPSC study has highlighted how sporadic APOE4 AD-iPSC astrocytes secrete higher levels of cholesterol (Table 5) and cholesteryl ester. When conditioned medium is added from these astrocytes to APOE4 sporadic AD-iPSC neurons, a higher secretion of Aβ42 and increased APP expression is observed [102]. In the case of AD, cholesterol secretion by astrocytes appears to facilitate the formation of lipid rafts in the plasma membrane, which results in increased Aβ42 production [102]. This is important since APP’s processing secretases, γ-secretase and β-secretase are both located in lipid rafts. Another study has found the mechanistic pathway potentially behind increased cholesterol biosynthesis. The study reported that AD-iPSC APOE astrocytes had dysregulated cholesterol metabolism due to lysosomal sequestration of cholesterol from the ER that resulted in increased cholesterol synthesis [103]. Further, APOE4 resulted in reduced levels of APOE and lipid transporters that decreased cholesterol efflux [103] leading to dysregulated cholesterol metabolism. Another study has shown that exposure of organoids to serum resulted in increased astrocytes in sporadic AD-iPSC organoids (Table 5). In addition, one study has clearly shown iPSC astrocytes can mimic differences in APP processing between different APP mutations. For e.g., iPSC astrocytes bearing the APP Swedish mutation increase beta-secretase cleavage of APP, whereas iPSC astrocytes bearing the APP^V717F^ mutation results in an increased Aβ42:40 ratio [104].

Only a handful of studies have attempted to recreate neuroinflammation events in AD using PSCs. Progress in the development of several different cell types from PSCs allows for more complex models to be produced, that model not only the brain parenchyma *per se*, but also the brain’s immune system. Complex cellular models are being developed that can recapitulate the innate immune response to a limited extent. Recently, triple cultures of healthy human PSC- microglia-like cells, astrocytes and cortical neurons have been established which can replicate a ratio of 1:11:20, microglia-like cells, astrocytes, and neurons, respectively. Under artificially simulated inflammatory challenge using lipopolysaccharide (LPS), the researchers were able to recapitulate an artificial neuroinflammatory state of the brain in vitro and secretion of classical inflammatory cytokines as well as observed secretion of C3 in microglia-like cells and astrocytes. The researchers then went on to model familial AD by using human APP^SWE^+/+ ESC-neurons co-cultured with WT astrocytes and microglia-like cells which demonstrated increased C3 and C1Q in cultures compared to triple cultures containing isogenic control neurons. However, no investigations were performed on the effect of increase C3 on synapse pruning, which could have helped recapitulate AD pathology.

**Table 5 cells-12-00420-t005:** Overview of glia dysfunction modelled in AD-iPSCs harboring either familial mutations, sporadic mutations or from late- onset sporadic AD patients.

Background	Cell type/s Analyzed	Observation	Reference
iPSC familial *APP* ^Swe^	iAstrocytes	Impaired Aβ intake, ↓ APP, ↑ Aβ40, 42, 38, → Aβ42:40 ↓ lipid endocytosis	[104] Fong
iPSC familial *APP* ^V717F^	iAstrocytes	→ impaired Aβ intake ↑ Aβ42:40, ↑Aβ42 → lipid endocytosis	[104] Fong
iPSC familial *PSEN1*	Astrocytes	↓ morphological heterogeneity, ↓ processes, ↓ release of IL-8, MCP-1	[105] Jones
iPSC sporadic *APOE4*	Astrocytes	↓Aβ uptake ↑ biosynthesis cholesterol	[72] Lin
iPSC sporadic *APOE4*	Astrocytes	↑ biosynthesis cholesterol	[102] Lee
iPSC sporadic *APOE4*	Astrocytes	↓ cholesterol efflux, ↓ biosynthesis ↑ cholesterol in lysosomes ↓ Aβ uptake ↑ cytokine secretion	[106] de Leeuw
iPSC sporadic *APOE4*	Astrocytes	↓ morphological heterogeneity, ↓ processes ↓ release of IL-8, MCP-1	[105] Jones
iPSC sporadic *APOE4*	Astrocytes	↑ cholesterol biosynthesis ↑ intracellular cholesterol ↓ cholesterol efflux → cholesterol esters	[103] Tcw
iPSC sporadic *APOE4*	Microglia-like cells	↑ cholesterol biosynthesis ↑ intracellular cholesterol ↓ cholesterol efflux	[103] Tcw
iPSC sporadic *APOE4*	Microglia-like cells	↓Aβ uptake Fewer, shorter processes	[72] Lin
iPSC down syndrome	Microglia-like cells in organoids	↑ synaptic pruning, ↑ phagocytosis, ↑ type 1-interferon signaling	[96] Jin

* iPSCs derived from confirmed AD patient. Abbreviations: iAstrocytes, induced astrocytes; NFT, neurofibrillary tangle. Arrows: ↑ increased, ↓ decreased, → no change.

## 5. Cell Signaling Alterations in iPSC Studies

Studies using bulk RNA sequencing continue to shed new light on cell signaling differences in AD-iPSC cell types. A comparative RNA sequencing study on human APOE4 iPSC neurons, astrocytes, and microglia-like cells show some overlap of dysregulated genes also observed in APOE sporadic AD patients [72]. Many genes relating to synapse function were observed in the iPSC neurons. In the case of iPSC microglia-like cells, most overlapping genes with patient brain were involved in immune system processes. Of interest, microglia-like cells had more dysregulated genes than those found in iPSC neurons and iPSC astrocytes. IPSC astrocytes showed altered genes in lipid metabolism [72]. In another study, bulk RNA seq was performed in iPSC organoids with either APOE3 or APOE4 genotypes; both derived from patients with confirmed AD [51]. Hundreds of genes were found to be differentially expressed in the APOE4 iPSC organoids when compared to APOE3 iPSC organoids with disruption in RNA metabolism observed [51], but an interpretation of the data is difficult, due to the lack of analysis of AD-specific pathways. 

With the advances of single cell RNA sequencing (scRNAseq), more details are to be learned on altered signaling in specific cell types in AD and will allow for better analyses of tri-cultures and organoids. Despite these advances, there are very few reported papers performing scRNAseq on AD-iPSC models. In one study, scRNAseq was performed on sporadic AD-iPSC organoids exposed to serum [55]. Neurons from these organoids had perturbed mitochondria function with increased expression of genes related to mitochondrial organization, electron transport chain, and oxidative phosphorylation [55]. In contrast, synapse-related genes were downregulated, confirming a loss of synapses after serum exposure [55]. In astrocytes, genes related to an enhanced immune response were identified from the same study in serum exposed organoids [55] which corroborated with an increased number of astrocytes observed, reflecting a reactive response to serum exposure. The method of scRNAseq has also been adopted as a tool to improve differentiation protocols of iPSCs. One study applied scRNAseq in combination with qPCR and phagocytosis assays to define an optimized protocol with media supplemented with TGB-β1. Comparative gene expression analyses were used to identify iPSC microglia-like cells that were most reminiscent of primary microglia [45].

Advances in proteomics has also meant that more proteins can be profiled from tissue and cellular samples. Proteomics also aids our understanding of dysregulating proteins within AD. One study used proteomics to identify protein differences between five control and five late-onset sporadic AD-iPSC organoids [107]. This led to the identification of 21 differentially regulated proteins across at least two of the patient-derived organoids, with 8 upregulated and 13 downregulated from a total of 1855 genes [107]. Upregulated genes were associated with transmembrane transporter activity and protein binding. Downregulated genes involved in ribosome regulation were downregulated in 3/5 organoids [107]. Proteins involved in GABAergic and dopaminergic synapses were downregulated as well as a decrease in oxidative phosphorylation [107]. These organoids failed to determine inflammation changes that were observed either in the processing of human AD brain samples or the role of complement that was also found perturbed. However, conservation in axonal development, mitochondrial dysfunction, and oxidative stress could be identified. That is, dysfunction within the neurons themselves could be observed. What this study also highlighted was the variability from sample to sample, indicating different mechanisms at play between sporadic AD patients. Proteomic studies using iPSCs have also helped to provide insight into early gene expression changes in AD. An analysis of protein changes in healthy iPSC neurons exposed to toxic forms of oligomeric Aβ highlighted a reduction in TDP-43, heterogeneous nuclear ribonucleoproteins, and coatomer complex I proteins and an increase of 20 S proteasome subunits and vesicle-associated proteins VAMP1/2 [108]. It is not clear from the literature yet that any of these iPSC studies have directly led to the identification of new gene targets that might modify the course of the disease. However, the use of AD-iPSCs in drug screening has been more promising when targeting major pathological hallmarks.

## 6. Using AD-iPSCs for Drug Discovery

The biggest advantage that human iPSCs bring to the AD field is their use and application in drug discovery. Scaling up iPSC cultures allows for the screening of large compound libraries. Protocols allow PSCs to differentiate into specific target human cell types, which is a significant advantage over other cell lines. Screening specific target cell types recapitulating the desired disease phenotype of interest means new compounds can be screened quickly for effects. Cellular screens such as iPSCs also reduce the number of animals required for testing as simple toxicity screens can also be performed. In the AD field, three notable drug screening studies using iPSCs have been published (Figure 2D). Two of these studies have focused on finding compounds that reduce Aβ and were published in 2017. In the first study, screening of iPSC iNeurons from a patient carrying the *PSEN1*^G384A^ mutation, led to the identification of several compounds that could reduce Aβ42 [109]. Of 1258 compounds tested, 11 compounds were found to lower Aβ42 of which several induced cell death in the cultures. Of a remaining six promising candidates, three were selected, bromocriptine, cromolyn, and topiramate, which were found to be most efficient when added as a cocktail (BCroT) to reduce Aβ in both sporadic and familial AD-iPSC iNeurons [109]. Independently of this discovery, another patented combinational therapy using cromolyn, but in combination with intal and ibuprofen (ALZT-OP1), entered clinical trials [110]. This drug is currently at the end of a Phase III clinical trial for patients with early stages of AD, with no posted results yet on the outcomes of this trial [110]. There are no indications to our knowledge that BCroT has entered clinical trials.

Another study published in the same year developed a drug screen using trisomy 21/Down syndrome iPSC-cortical neurons to also identify modulators of APP processing. This iPSC model contains an extra copy of APP which leads to early onset AD and all Aβ peptides are produced more than control cell lines. The aim in this study was to identify compounds that could reduce the longer and more toxic form of Aβ, Aβ42. The endpoint was to determine an increased shift in the Aβ38:42 and Aβ40:42 ratios. From 1200 screened compounds, an initial 55 compounds were identified in a primary screen. Following validation experiments, two compounds were found to increase the Aβ38:42 by lowering Aβ42 (compounds not disclosed). Additional compounds were studied including, (R)-flurbiprofen, E2012, abamectin, and ivermectin. The two latter are members of the avermectin class of macrocyclic lactones, and are currently used to treat parasitic infections, but do not cross the BBB [111]. Subsequent analyses of other avermectins in this study showed similar results [111]. The study attempted to identify the mechanism of action of avermectins on modulating APP processing. The mechanism appeared to be independent of direct actions of the core γ-secretase complex and did not appear to influence Aβ processing via known pharmacological targets [111]. None of the avermectins have made progress to clinical trials for AD to the best of our knowledge, likely given the inability of these drugs to cross the BBB. E2012 had been discovered in earlier studies and was tested in a Phase I clinical trial, but was not pursued further [112]. Similarly, (R)-flurbiprofen was discontinued after testing until Phase III clinical trials in 2008, due to lack of efficacy [113]. 

Human iPSC neurons have also been used in a compound screening study to identify regulators of pTau. In this study, more than 1684 compounds were screened to identify 96 compounds that could reduce pTau accumulation in familial APP^Dp^ AD-iPSC neurons [114]. Six of these hits were microtubule-interacting compounds, whilst four others were statins that inhibited cholesterol synthesis [114]. This study focused on the statin hits and went on to identify the mechanism of action in regulating pTau. The tested statins were found to inhibit an early rate limiting step in cholesterol synthesis that converts HMG-CoA to mevalonate (MVA) in a reversible-dependent manner [114]. However, this was not directly linked to the modulation of pTau levels. The study identified the statins regulated either the cholesterol–synthetic branch of the MVA pathway or activated a neuron specific enzyme that converts cholesterol to 24-hydroxycholesterol and found that cholesterol esters regulated pTau [114]. This regulation of pTau was found to be conserved across different patient and control lines [114]. Further testing of some of these statins revealed toxicity in astrocytes; however, one compound, efavirenz, did not display such toxicity [114]. In an independent study, a 6-month treatment of efavirenz resulted in behavioral improvements in 5XFAD mice [115] and this compound went on to be tested in a Phase I clinical trial [116]. This trial ended early 2022, but the results have not been revealed. Regardless of the outcomes from clinical trials, these reports highlight how AD-iPSC neurons are an excellent source for discovering novel compounds for the treatment of AD. They can also be used for discovering mechanisms by which drugs act on the neurons and for determining drug toxicity.

In addition to these promising studies, many other iPSC studies have tested individual compounds on their ability to modulate Aβ. A couple of examples are highlighted here. Both β and γ-secretase inhibitors have been long revered as a possible target for correcting impaired APP processing. One study by Raja et al. was able to successfully reduce total Aβ levels and levels of pTau in organoids derived from familial forms of AD after 30 and 60 days of treatment with combined exposure to a γ-secretase inhibitor and a BACE-1 β-secretase inhibitor [54]. Interestingly, more striking effects were reported with longer treatment (60 days) [54]. The γ-secretase and β-secretase inhibitors that have been taken into clinical trials have not fared well, due to adverse side effects or lack of efficacy in patients [117]. In another study, one small molecule that had been identified to bind specifically with APOE4 and render changes to it structurally was tested in sporadic AD-iPSC APOE4-neurons [73]. The study demonstrated that this molecule effectively decreased APOE4 fragment levels, reduced pTau, and decreased Aβ42:40 ratios [73]. It will be interesting to follow the path of this molecule and many others that arise from testing using AD-iPSC screens.

Organoids also have great promise in screening compounds of interest for AD, although there are no published reports yet revealing outcomes from testing of these using compound screens to the best of our knowledge. Since 2018 there have been no further reports on compound screening on AD-iPSCs. This is largely since industry has realized the great potential of using AD-iPSC neurons and iPSC organoids for drug discovery [118]. In one published study, >1000 human iPSC organoids from sporadic AD patients were used to develop a high-content screening system which was used along with exclusion criteria and mathematical modeling of gene-signaling pathways, to identify a small cohort of FDA-approved drugs with promise for treating AD [74]. However, the paper did not share the results of the testing of these drugs. 

Other neural cell types are also being targeted in more recent publications to account for changes in the immune system in AD. Microglia are an attractive cell type to target, given they express TREM2 (a risk gene of AD). They also play a major role in the uptake and clearing of Aβ, as well as mediate synapse pruning and contribute to neuroinflammation [119]. An iPSC microglia cell screening system has recently been developed that incorporates CRISPR machinery tools that allow for the potential to screen for differences in microglial cell states [42]. This is expected to be a useful screen for identifying novel compounds that treat immune dysfunction in AD. Another automated platform has been designed which includes iPSC neurons, astrocytes, and microglia-like cells [120]. In this article, the researchers investigated the role of microglia in the presence of soluble toxic Aβ42 [120]. Time-lapse imaging was able to show how microglia-like cells engulfed the toxic Aβ42 before plaques formed, suggesting microglia-like cells may play an important role in plaque formation by exocytosing them as plaque structures [120]. This study used a complex iPSC culture system to generate new hypotheses on plaque formation.

In sum, breakthroughs in drug screening using iPSCs have been published, but there are likely many more promising compounds that have been discovered and are not known to the public. The potential of iPSCs in early-stage drug discovery for AD is in no doubt huge. Drug discovery therefore remains a top attribute for using AD-iPSC models, largely since they recapitulate the major hallmarks remarkably well from the mutations they harbor or from the patients they are derived from. This raises the possibility of AD-iPSCs being used in the future for facilitating the development of customized personal medicine approaches for treating AD.

## 7. Caveats and Use of iPSC Research

Most studies have performed modeling of AD using 2D single neuron cultures, which provide useful, but limited information on disease mechanisms. The main neuron subtype studied is cortical neurons. Often, these neurons do not reflect the mature morphology observed in the brain and require mature astrocytes in co-culture to acquire a mature phenotype [121]. 2D single neuron cultures remain at best, suitable for studying early disease mechanisms. Further, neuronal degeneration is not widely reported in iPSC neurons, which also suggests these models cannot recapitulate later stages of the disease. The AD-iPSC field could better contribute to our understanding of AD by studying specific vulnerable cell subtypes and studying more on the less commonly studied pathologies discussed in Section 4.3. Recent developments have shown that entorhinal neurons and certain astrocyte subtypes are more vulnerable than others. With new protocols in place e.g., for a specific and vulnerable entorhinal neuron such as the stellate cell [46], more light could be shed onto the early mechanisms associated with AD. Whilst APP processing and tau phosphorylation have been well studied, many pathologies are neglected when studying AD-iPSC models. Studying less known pathological hallmarks could certainly add more knowledge on disease pathogenesis. AD-iPSCs could also be better used for studying potential environmental risk factors. Implicated dietary deficiencies, inflammation, microbiome-related perturbations, and infections for example could be investigated using these models to learn more about how the environment and diet contribute to the disease. 

Given the complexity of AD pathology, it is more apparent that more complex cellular models are needed when studying the disease in vitro. The field is moving away from single cell neuron cultures, and research in the field has already highlighted an emergence of more complex cell systems. The emergence of 2D triple cultures containing astrocytes and microglia results in more mature neuron phenotypes and allows for events related to the immune system to be studied. 2D co- and triple cultures are a step in the right direction for studying the disease. However, there remains missing elements in these cultures, including the vasculature. There is also a lost perspective of the role of peripheral changes that contribute to brain health, such as metabolic changes and systemic inflammation that may have an effect or contribute to disease progression. In the case of more complex organoids, the quality of organoid production appears to vary a great deal. Caution is therefore required when evaluating pathology and neuronal activity in organoids. Some research groups carefully select only the best quality organoids for analyses [55]. Large variation in cell types have been found within organoids and differences in neuronal activity discovered within the same parental lines [100]. Therefore, improvements in standardization of organoid production are required and a careful evaluation of organoids in papers is needed, to convince readers of reproducible, robust data. 

Human iPSC neurons recapitulate most, but not all features of AD. One caveat is that iPSC neurons secrete less four-repeat forms (4R) of tau (found in the adult brain) than observed in the brain and CSF, and more 3R (fetal brain version) isoforms [122]. It is likely that altered forms of tau do have some impact on the outcomes when studying tau phosphorylation in iPSC models. It remains a challenge to produce mature neuronal phenotypes from iPSCs. However, in general, longer iPSC cultures result in more robust AD phenotypes due to the ability for neurons to further mature. Another caveat is the recent discovery that the structure of Aβ42 fibrils secreted in culture differ from those found in the AD brain. This finding is unlikely to have a major impact on most APP processing studies in the short term, or on drug discovery using AD-iPSCs, as proof-of-concept studies already show great promise in using AD-iPSC drug screens. However, careful consideration of this new finding is required for more detailed studies on fibrillar forms and seeding of Aβ fibrils using in vitro cellular models. 

IPSC culture probably remains the most expensive cell culture that exists today. This is a major caveat in the field. The costs of iPSC experiments are directly attributable to the high cost of commercial media and associated products and matrices required to cultivated them, as well as the extensive media changing rituals required to maintain healthy iPSC lines. More effort is required to bring the costs associated with iPSC culture down, since these costs limit study design and experimental capacity for many labs. One way to circumvent this could be through open publications of specific media recipes and the development of new formulations containing cheaper components. 

In general, AD-iPSCs can model neuronal dysfunction well and recapitulate more complex brain tissue to study the role of the immune system (with the addition of microglia) and the supportive role of astrocytes. AD-iPSCs can provide insight into cellular changes, mechanisms and may also be useful for identifying biomarkers for early disease, albeit no new biomarkers have yet been identified. More research using AD-iPSC co- and tri-cultures and organoids will be key to providing new data on mechanistic interactions between dysfunctional glia and neurons. Despite the caveats of 2D neuron cultures, these have been successfully used to identify novel compounds that modulate Aβ and tau. AD-iPSCs provide an excellent model for testing newly identified compounds/drugs for reversal of AD pathology as a first step in testing of toxicity and efficacy. Together, the different forms of AD models remain worthy tools for studying AD, when access to human tissue is limited. 

## 8. Conclusions

In sum, over the course of 15 years, modeling of AD using iPSCs has progressed from simple 2D cultures of cortical neurons to more complex 3D models that now in many systems include astrocyte and microglia-like cell types. APP processing has been well studied and AD-iPSC models reproduce dysregulated Aβ well. Tau phosphorylation has also been well studied and shows dysregulation similar, but not always identical to that described in patients. There are only a few studies reporting the presence of plaques and tangles in AD-iPSC models, largely since more complex models of AD using iPSCs are just emerging. AD-iPSCs have shown greatest promise in discovery of new compounds that modulate both Aβ and tau. There is still more to be learned from AD-iPSCs using the advances in gene technologies such as scRNAseq and proteomic profiling. Modeling of LATE-AD and many other mutations in familial AD are required. This may provide further insight into the complexity and individuality of patient pathology. Finally, AD-iPSCs may be helpful in development of patient-customized therapies in the future given our understanding of the variability in the pathological processes of the disease.

## Figures and Tables

**Figure 1 cells-12-00420-f001:**
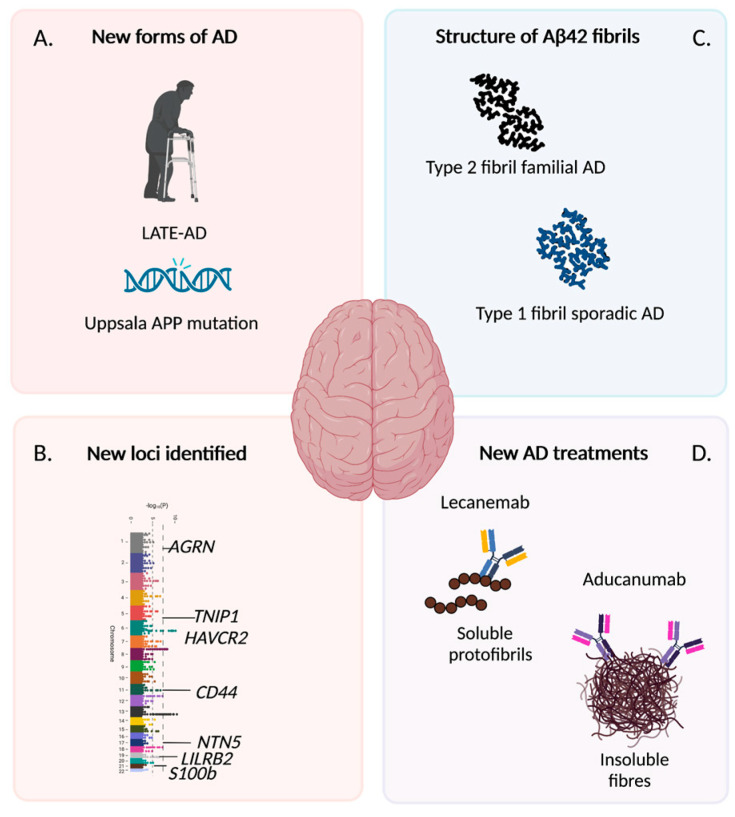
Highlighted major breakthroughs in the Alzheimer’s disease (AD) field over the last decade. New forms of AD have been identified including a sporadic form called LATE [2] and a familial form referred to as the Uppsala APP mutation [3] (**A**). New gene loci associated with development of AD have been identified, with a few of the recent loci from [4] and [5] shown here (**B**). Researchers have identified the structural insights of amyloid beta (Aβ) showing fibrils differ between familial and sporadic AD [6] (**C**). Progress has recently been made in finding treatments for AD including Lecanemab and Aducanumab, which have recently been approved and marketed in the US. Both target the removal of Aβ but have different preferences in their ability to remove either soluble protofibrils or insoluble fibers (**D**). Created with BioRender.com.

**Figure 2 cells-12-00420-f002:**
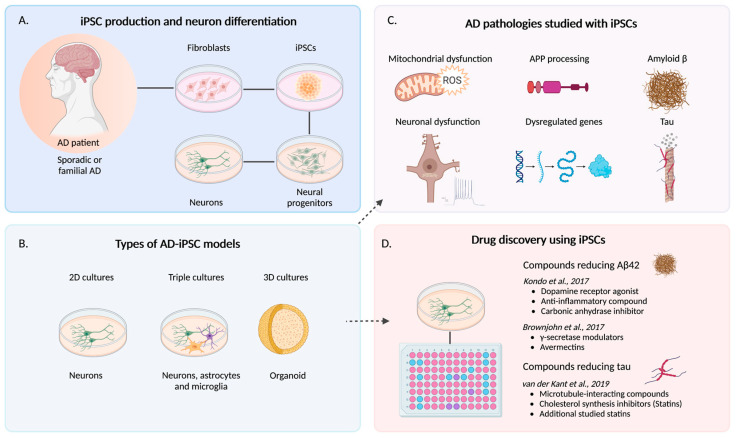
Modeling of Alzheimer’s disease using iPSCs. Human iPSCs are produced from fibroblasts and differentiated into neurons (**A**) that can be cultured in simple 2D conditions, triple cultures, or in 3D as organoids (**B**). Human AD-iPSCs have been used to study several different pathologies including amyloid precursor protein (APP) processing and increased amyloid beta 42 (Aβ42), tau phosphorylation and neuronal dysfunction (**C**). AD-iPSCs have also been used to screen and find candidate compounds that can reduce Aβ42 and tau (**D**). Created by BioRender.com.

**Figure 3 cells-12-00420-f003:**
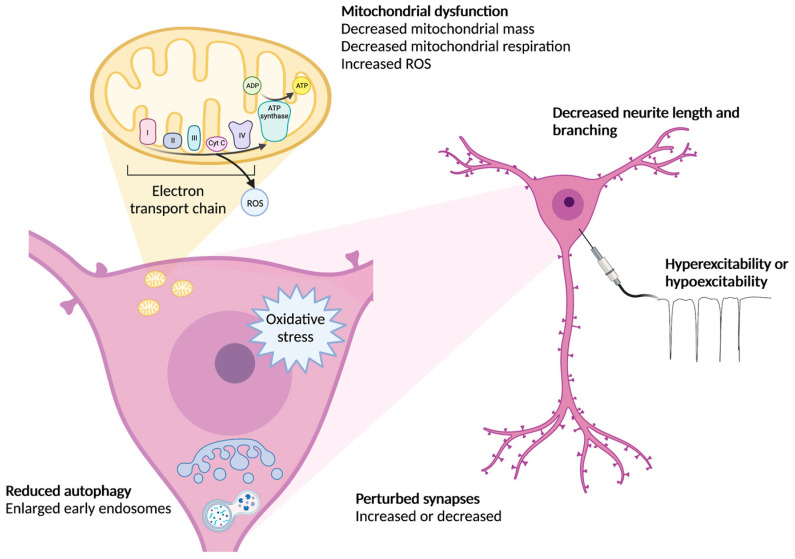
Perturbed organelle and neuron function have been reported in both familial and sporadic AD-iPSC neurons. Mitochondrial dysfunction, reduced autophagy, perturbed numbers of synapses, decreased neurite lengths and branching as well as perturbed excitability have all been reported and reflect known phenotypes shown in the AD brain. However, not all iPSC neurons show these phenotypes and variabilities between cell lines and risk genes/mutations have been noted. Created with BioRender.com.

**Table 1 cells-12-00420-t001:** Overview of amyloid beta (Aβ) pathology modelled in AD-iPSCs harboring either familial mutations, sporadic mutations or from late-onset sporadic AD patients.

Background	Cell type/s Analyzed	Observation	Reference
ESCs familial *APP* ^SWE^	Cortical neurons	↑ Aβ38, ↑ Aβ40, ↑ Aβ42	[41] Guttikonda
iPSC familial *APP* ^SWE/V717I^	Cortical neurons	↑ Aβ40, ↑ Aβ42	[49] Park
iPSC familial *APP* ^SWE, A692G^	Cortical neurons	↑ total Aβ, ↑ Aβ42 → Aβ42:40	[56] Kwart
iPSC familial *APP* ^V717G^	Cortical neurons	↑ Aβ42:40 ↑ Aβ42	[56] Kwart
iPSC familial *APP* ^V717I^	Cortical neurons	↑ Aβ38:40 ↑ Aβ42:40	[62] Arber
iPSC familial *APP* ^V717I^	Cortical neurons	↑ Aβ42:40 ↑ Aβ42, ↑ Aβ38	[63] Muratore
iPSC familial *APP* ^V717I^	Cortical neurons	→ intracellular Aβ oligomers	[64] Kondo
iPSC familial *APP* ^Dp^	Cortical neurons	↑ Aβ40	[65] Israel
iPSC familial *APP* ^E693delta^	Cortical neurons	↑ intracellular Aβ oligomers	[64] Kondo
iPSC familial *APP* ^V717I^	Organoids	↑ Aβ42:40 ↑ Aβ42:38	[62] Arber
iPSC familial *APP* ^Dp^	Organoids	Aβ aggregates, → Aβ42:40 ↑ Aβ42, ↑ Aβ40	[54] Raja
iPSC familial *PSEN1* ^A246E, N141I^	Cortical neurons	↑ Aβ42:40 → Aβ40	[66] Yagi
iPSC familial *PSEN1* ^A246E^	Cortical neurons	↑ Aβ42:40	[67] Mahairaki
iPSC familial *PSEN1* ^Y115H, int4del, M139V, M146I, R278I^	Cortical neurons	↑ Aβ38:40 ↑ Aβ42:40	[62] Arber
iPSC familial *PSEN1* ^M146V, L166P, M233L, A246E^	Cortical neurons	↓ total Aβ, ↑ Aβ42 ↑ Aβ42:40	[56] Kwart
iPSC familial *PSEN1* ^V89L, L150P^	Cortical neurons	↑ Aβ42:40 ↑ Aβ42, ↑ Aβ40	[68] Ochalek
iPSC familial *PSEN1* ^A246E^	Organoids	Extracellular Aβ ↑ Aβ42:40	[69] Gonzalez
iPSC familial *PSEN1* ^Y115H, int4del, M139V, M146I, R278I^	Organoids	↑ Aβ42:40 ↑ Aβ42:38	[62] Arber
iPSC familial *PSEN1* ^A264E^	Organoids	Aβ aggregates	[54] Raja
iPSC familial *PSEN1* ^M146I^	Organoids	→ Aβ42, → Aβ40	[54] Raja
iPSC familial *PSEN2* ^N141I^	Basal forebrain cholinergic neurons	↑ Aβ42:40 ↑ Aβ	[70] Ortiz-Virumbrales
iPSC sporadic *APOE3 **	Organoids	↑ Aβ40, ↑ Aβ42	[51] Zhao
iPSC sporadic *APOE3/4*	Basal forebrain cholinergic neurons	↑ Aβ42:40	[71] Duan
iPSC sporadic *APOE4*	iNeurons	↑ Aβ42 → Aβ40	[72] Lin
iPSC sporadic *APOE4*	Cortical neurons	↑ Aβ42 ↑40 ↑ sAPP	[73] Wang
iPSC sporadic *APOE4*	Organoids	↑ Aβ40, ↑ Aβ42	[51] Zhao
iPSC sporadic *APOE4*	Organoids	↑ Aβ42	[74] Park
iPSC sporadic *APOE4*	Organoids	↑ Aβ42, ↑ Aβ40	[75] Huang
iPSC sporadic *	Cortical neurons	→ Aβ42:40 ↑ Aβ42, ↑ Aβ40	[68] Ochalek
iPSC sporadic *	Cortical neurons	↑ Aβ40	[65] Israel
iPSC sporadic *	Cortical neurons	↑ Aβ42, → Aβ40	[76] Balez
iPSC sporadic *	Cortical neurons	↑ intracellular Aβ in 1 of 2 lines	[64] Kondo
iPSC sporadic	Organoids	→ Aβ40, no amyloid aggregates	[55] Chen
iPSC sporadic + serum	Organoids	↑ Aβ40, Aβ aggregates	[55] Chen

* iPSCs derived from confirmed AD patient. Arrows: ↑ increased, ↓ decreased, → no change.

**Table 2 cells-12-00420-t002:** Overview of tau pathology modelled in AD-iPSCs harboring either familial mutations, sporadic mutations or from late-onset sporadic AD patients.

Background	Cell type/s Analyzed	Observation	Reference
iPSC familial *APP* ^V717I^	Cortical neurons	↑ total tau, pTau^Ser262^	[63] Muratore
iPSC familial *APP* ^Dp^	Cortical neurons	↑ pTau^Ser231^	[65] Israel
iPSC familial *APP* ^Dp^	Organoids	↑ pTau^Ser231, Thr181^	[54] Raja
iPSC familial *PSEN1* ^V89L, L150P^	Cortical neurons	↑ pTau^Ser262, Ser396, AT8, Thr181, Ser400/Thr403/Ser404^	[68] Mahairaki
iPSC familial *PSEN1* ^A264E^	Organoids	↑ pTau^Ser231^	[54] Raja
iPSC familial *PSEN1* ^M146I^	Organoids	→ pTau^Ser231^	[54] Raja
iPSC familial *PSEN1* ^A246E^	Organoids	↑ pTau, NFTs	[69] Gonzalez
iPSC sporadic *	Cortical neurons	↑pTau^Ser262, Ser396, AT8, Thr181, Ser400/Thr403/Ser404^	[68] Mahairaki
iPSC sporadic *	Cortical neurons	↑ pTau^Ser396, Thr181^	[65] Israel
iPSC sporadic *APOE3 **	Organoids	→ pTau	[51] Zhao
iPSC sporadic *APOE4*	Cortical neurons	↑ pTau	[73] Wang
iPSC sporadic *APOE4*	Organoids	↑ pTau	[51] Zhao
iPSC sporadic *APOE4*	Organoids	→ secreted total tau, ↑ pTau	[74] Park
iPSC sporadic *APOE4*	Organoids	↑ pTau	[75] Huang
iPSC sporadic + Serum	Organoids	↑ pTau	[55] Chen

* iPSCs derived from confirmed AD patient. Abbreviations: NFT, neurofibrillary tangle. Arrows: ↑ increased, ↓ decreased, → no change.

**Table 3 cells-12-00420-t003:** Overview of organelle pathology modelled in AD-iPSC neurons and glia harboring either familial mutations, sporadic mutations or from late-onset sporadic AD patients.

Background	Cell type/s Analyzed	Observation	Reference
iPSC familial *APP* ^V717L^	Cortical neurons	↓autophagy, ↓mitophagy	[7] Fang
iPSC familial *APP* ^swe, A692G, V717G^	Cortical neurons	RAB5+ early endosome enlargement	[56] Kwart
iPSC familial *APP* ^Dp^	Cortical neurons	RAB5+ early endosome enlargement	[65] Israel
iPSC familial *APP* ^E693delta^	Cortical neurons	↑ ER stress and oxidative stress	[64] Kondo
iPSC familial *APP* ^V717I^	Cortical neurons	→ ER stress and oxidative stress	[64] Kondo
iPSC familial APP ^Dp^	Organoids	↑ large endosomes	[54] Raja
iPSC familial *PSEN1* ^L166P^	Cortical neurons	RAB5+ early endosome enlargement	[56] Kwart
iPSC familial *PSEN1* ^P117L^	Basal forebrain cholinergic neurons	Mitochondrial dysfunction, ↓ mitochondrial membrane potential, ↑ ROS, ↑ superoxide	[83] Oka
iPSC familial *PSEN1* ^A246E^	Cortical neurons	↓autophagy ↑ autophagic vacuoles ↑ lysosomal biogenesis	[92] Martín-Maestro
iPSC familial *PSEN1* ^M146I, A264E^	Organoids	↑ large endosomes	[54] Raja
iPSC sporadic *APOE4*	Cortical neurons	↓autophagy, ↓mitophagy	[7] Fang
iPSC sporadic *APOE4*	iNeurons	↑ early endosomes	[72] Lin
iPSC sporadic AD *	Cortical neurons	↑ ER stress and oxidative stress	[64] Kondo
iPSC sporadic AD *	iNeurons	Mitochondrial dysfunction ↑ ROS in some lines ↑ oxidative phosphorylation	[84] Birnbaum
iPSC sporadic AD *	Cortical neurons	RAB5+ early endosome enlargement	[65] Israel
iPSC sporadic AD *	Cortical neurons	↓ Cox Vmax ↓ mitochondrial mass ↓ mitochondrial respiration ↓ mitochondrial membrane potential ↓ superoxide	[85] Flannagan
iPSC sporadic AD *	Organoids	↓ COX Vmax	[85] Flannagan
iPSC sporadic AD *	Astrocytes	→ COX Vmax ↓ mitochondrial respiration ↑ mitochondrial membrane potential → superoxide	[85] Flannagan
iPSC sporadic *SORL1* KO	Neurons	early endosome enlargement	[88] Knupp
iPSC sporadic *SORL1* KO	Microglia	→ in endosome	[88] Knupp

* iPSCs derived from confirmed AD patient. Abbreviations: iNeurons, induced neurons; COX, cytochrome oxidase; ER, Endoplasmic reticulum; KO, knockout; ROS, reactive oxygen species. Arrows: ↑ increased, ↓ decreased, → no change.

**Table 4 cells-12-00420-t004:** Overview of neuronal dysfunction identified in AD-iPSC neurons harboring either familial mutations, sporadic mutations, or from late-onset sporadic AD patients.

Background	Cell type/s Analyzed	Observation	Reference
iPSC familial *APP* ^Swe^	Cortical neurons	Hyperexcitability → resting membrane potential → action potential firing threshold ↓ Neurite length, branching ↑ excitatory synaptic activity	[101] Ghatak
iPSC familial *PSEN1* ^deltaE9^	Cortical neurons	Hyperexcitability → action potential firing threshold ↓ Neurite length, branching ↑ excitatory synaptic activity	[101] Ghatak
iPSC familial *PSEN1* ^M146V^	Cortical neurons	Hyperexcitability → action potential firing threshold ↓ Neurite length, branching ↑ excitatory synaptic activity	[101] Ghatak
iPSC familial *PSEN2* ^N141I^	Basal forebrain cholinergic neurons	↓ excitability	[70] Ortiz-Virumbrales
iPSC sporadic *APOE3/4* *	Cortical neurons	↓ synaptic density	[95] Wang
iPSC sporadic *APOE3/3* *	Cortical neurons	↓ synaptic density	[95] Wang
iPSC sporadic *APOE4*	iNeurons	↑ synapses	[72] Lin
iPSC sporadic *APOE4*	Organoids	↑ cholesterol and lipid droplets	[75] Huang
iPSC sporadic *APOE4*	Organoids	↓ synapses	[51] Zhao
iPSC sporadic *	Cortical neurons	↓ Neurite length, hyperexcitability	[76] Balez
iPSC sporadic + serum	Organoids	↓ synapses	[55] Chen

* iPSCs derived from confirmed AD patient. Abbreviation: iNeurons, induced neurons. Arrows: ↑ increased, ↓ decreased, → no change.

## Data Availability

Not applicable.

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
