# Peer review of "The Breakthroughs and Caveats of Using Human Pluripotent Stem Cells in Modeling Alzheimer’s Disease"

_cells, 2023, doi:10.3390/cells12030420_

Round 1
Reviewer 1 Report
This paper describes the role of pluripotent stem cells in the study of AD. The manuscript is well written and gives plenty of information on the use if iPS cells in AD. I have a few minor comments.
1) Section 2 seemed a little out of place in context with the whole manuscript. While the rest of the manuscript details how iPS cells is a beneficial tool in the study of AD, Section 2 did not transition well into the rest of the sections.
2) It would be great of the authors could give caveats of other models more extensively and then discuss the uses of iPS cells in the study of AD
Author Response
We thank the reviewer for the promising comments and address the two points that have been made which we hope lead to better clarity, flow and improve understanding.
- We agree that we failed to transition the manuscript well between section 1 and the rest of the sections. We have modified the last paragraph of Section 1 to better transition the review from the breakthroughs in the AD field, to the rest of the paper on iPSCs. See lines 160-169.
- We have extended section 7 by half/3/4 page to give a better description of caveats of other models (we elaborate on 2D single and co-triple cultures), and end the section with a new paragraph on the use of iPSCs for studying AD. See lines 528-532, 545-550 and 572-580 for the new text added.
Reviewer 2 Report
The manuscript from Sahlgren and Hall titled “The breakthroughs and caveats of using human pluripotent stem cells in modelling Alzheimer’S disease” is a short review describing the current knowledge regarding the recent advances obtained by modeling Alzheimer’s disease (AD) using human induced pluripotent stem cells (iPSC).
The review starts giving a short summary of selected recent findings regarding the pathology and treatment of AD. The authors then give a general overview of the different studies that used iPSCs to model AD to go in the details and describe how well iPSC models reproduce typical pathological hallmarks of AD. In addition, the review addresses the studies performed so far that use iPSCs to screen new substances for drug discovery and finally goes into some of the caveats of the iPSC research and open questions for the future.
All in all this is a very comprehensive and well written review which gives a clear and objective view about the strengths and problems of modeling AD with iPSCs. Taken together I think that this review is a very nice addition for the field. However, I have two comments that need to be addressed before publications:
1) The major criticism that I have regarding this review is about the figures. I think that in their current version the figures are not very informative and do not add much to the text. In additions, the figures are not well integrated into the text. For example figure 1: the single panels could be labelled (A, B …etc) so that they can be better referred to in the text. Also for instance the graph showing the new gene loci identified as it is is essentially useless…better would be to make it bigger and instead of writing the genes label them in the graph. Also the images chosen for the Aβ structure, the new forms of AD and the brain in the middle don’t really add information…
2) It may be nice to have also a very short description of the different in vitro preparations the authors mention in the text: 2D, 3D cultures and organoids.
Minor point:
1) There are a few spelling mistakes that should be corrected (e.g. at line 119 “has” should be changed into “have”) so please read it one more carefully through.
Author Response
We thank the reviewer for the positive review and we address their comments here.
- We have worked hard to improve the figures and incorporate them better into the review. This has resulted in a redesign of Figure 1 and a new Figure 2, and removal of the old Figure 3.
Figure 1 now has four sub-figures and has been better incorporated into Section 1 of the review.
Figure 2 provides an overview of iPSC production, types of AD-iPSC models, AD pathologies studied and drug discovery using iPSCs. We have integrated the sub-figures well into the text.
Figure 3 we retain as we believe it gives a good overview of the different neuronal dysfunctions found in AD-iPSC neurons.
- We have now added a clear succinct description of 2D, and 3D cultures. See lines 186-192
- We have carefully read through the text to correct for minor spelling errors.